# Effect of Extended Photoperiod on Performance, Health, and Behavioural Parameters in Nursery Pigs

**DOI:** 10.3390/vetsci10020137

**Published:** 2023-02-09

**Authors:** Friso Griffioen, Marijke Aluwé, Dominiek Maes

**Affiliations:** 1Unit of Porcine Health Management, Department of Internal Medicine, Reproduction and Population Medicine, Faculty of Veterinary Medicine, Ghent University, 9820 Merelbeke, Belgium; 2Flanders Research Institute for Agriculture, Fisheries and Food, Animal Knowledge Centre (ILVO), 9820 Merelbeke, Belgium

**Keywords:** piglets, nursery, light, photoperiod, performance, health, welfare

## Abstract

**Simple Summary:**

Providing more light in the first days after weaning is a common practice in pig husbandry. This supposedly stimulates feed intake and performance of (smaller) piglets after weaning. Currently, the literature does not agree on whether this is beneficial. Therefore, the effects of extending the lighting schedule on nursery pig performance, health, and behaviour during the entire nursery period were studied. No beneficial effects were seen on performance and behaviour of the pigs. Furthermore, piglets that were exposed to more light showed worse fecal consistency scores on days 7 (LONG: 21 vs. SHORT: 10; *p* = 0.039), 11 (LONG: 40 vs. SHORT: 14; *p* = 0.002), 21 (LONG: 21 vs. SHORT: 8; *p* = 0.008), and 25 (LONG: 26 vs. SHORT: 11; *p* = 0.015). It could not be excluded that this difference was due to infection, but there was no significant increase in mortality. This study provides evidence that leaving the lights on for the first days after weaning is not beneficial but should be repeated on more farms to get definitive answers.

**Abstract:**

Lighting influences the circadian rhythm and physiology of animals. Yet, the influence of light on nursery pigs is not fully understood and results remain controversial. The present study investigated the effects of a prolonged photoperiod on the performance, health, and behaviour of nursery pigs. This study was conducted in one farm and included 288 hybrid nursery pigs. Long (LONG) and short (SHORT) photoperiod animals were exposed to either 16 or 8 h of light per day, respectively. Performance, health, welfare, and behavioural parameters were monitored during a nursery period of five weeks. Short photoperiod piglets tended to have higher weights at the end of the nursery (LONG: 21.59 vs. SHORT: 22.19 kg; *p* = 0.064) and higher average daily gain (LONG: 385 vs. SHORT: 403 g/day; *p* = 0.063) compared to the long photoperiod piglets. The LONG piglets had significantly higher fecal consistency scores (0–100) than the SHORT piglets on days 7 (LONG: 21 vs. SHORT: 10; *p* = 0.039), 11 (LONG: 40 vs. SHORT: 14; *p* = 0.002), 21 (LONG: 21 vs. SHORT: 8; *p* = 0.008), and 25 (LONG: 26 vs. SHORT: 11; *p* = 0.015). The LONG piglets tended to have a slightly higher incidence of aggressive behaviour (LONG: 2.49% vs. SHORT: 2.36%; *p* = 0.071). No significant differences were found for the remaining parameters (*p* > 0.1). Under the present conditions, lengthening the photoperiod during the nursery period did not significantly improve the performance, health, and welfare of the pigs.

## 1. Introduction

Weaning brings about multiple changes for piglets. First, the piglets are taken away from their mother and are moved into a new environment with unfamiliar pen-mates. Second, they cannot suckle milk anymore and are often switched to a solid diet. These changes decrease feed intake during the first days after weaning, resulting in growth retardation [1]. The lower feed intake causes shortening of the intestinal villi and increases mitotic activity in villus crypts (crypt hyperplasia). These morphological changes reduce the efficiency of nutrient uptake, resulting in a vicious circle until feed uptake reaches normal levels. All these events lead to a higher susceptibility for intestinal infections and poor performance post weaning [2,3]. Modifications of the light schedule have been proposed as a possible way to improve performance after weaning [4,5,6,7,8]. Theoretically, a longer photoperiod facilitates piglets to have more and/or better access to feed. However, studies are scarce and results controversial [9,10,11].

Mammalian circadian rhythms are usually around 24 h in length and are influenced by several “Zeitgeber”, i.e., environmental factors. High temperatures move the activity peaks to the colder periods of the day, while permanent exposure to light induces a loss of the cyclic activity pattern [12,13]. Pigs are diurnal animals, meaning that they are most active during the day and sleep during the night [12,13,14]. During the day, both wild and commercial pigs have a diurnal rhythm with two main activity peaks, one occurring at dawn and one at dusk. Wild female pigs are active throughout the entire day, whereas males show short outbursts of activity interspersed with long periods of rest [12,14,15]. The same is true for the domesticated pig. Light has been shown to be a powerful tool to alter circadian rhythms in mammals [13]. It influences hormone secretion in male wild boars and domestic pigs [16]. During the summer months, a clear influence of light and temperature is noted on the reproductive performance of domesticated sows and boars [16,17,18,19,20]. Manipulating the diurnal rhythms of pigs using light might increase the number of activity peaks in a day, thus holding the key to successfully decreasing the post-weaning gap. The importance of light is determined by three main characteristics, namely the color or wavelength (nm), intensity (lux), and photoperiodicity (h). Minimal intensity as well as photoperiod are determined by legislation in the EU, as pigs have to be exposed to a light intensity of 40 lux for at least eight hours per day [21].

The effect of photoperiod on the production of cattle and poultry has been shown previously [22,23,24,25]. Furthermore, a profound influence of photoperiod on the fertility of sows and boars has been described [16,17,18,19,20]. Conversely, little is known about the effects of light on weaned piglets. Several studies have shown a positive influence of increasing photoperiod. Bruininx et al. (20002b) showed an increase in daily feed intake (ADFI) and average daily gain (ADG) for piglets that were exposed to a prolonged photoperiod of 23 h of light and 1 h of darkness (23L1D), compared to a control group (8L16D) [6]. Niekamp et al. (2007) showed that an increased photoperiod (16L8D) and weaning age (28 vs. 14 days of age) improved ADG, end weight, and several immunity parameters [7]. Kluivers-Poodt et al. (2018) exposed piglets, kept under either TL- or LED-lighting, to an extended photoperiod (16L8D) [8]. The results showed that the feed conversion ratio (FCR) and ADFI of piglets that were kept under LED-light were higher during the first 14 days, compared to piglets that were kept under TL-light. However, alternating the lighting schedule by switching between high (150 lux) and low (40 lux) intensities throughout the day, improved FCR and ADG compared to the TL-group [8]. Contrarily, multiple studies have found no or a negative effect of prolonging the photoperiod. Researchers investigated whether lighting conditions mimicking those of the farrowing room might positively influence the performance of weaned piglets. Treatment piglets were exposed to pascal red lighting at night, emulating the heat lamp in the farrowing unit. Piglets showed higher ADFI and FCR during the first 7 days of the trial. This finding was not observed during the entire period [9]. Differences in photoperiod may also affect behaviour and welfare. Gomes et al. (2018) showed a decline in feed:gain and higher cortisol levels during the first 14 days when piglets were exposed to 23L1D compared to 12L12D [10]. Lastly, the quality of light might influence the occurrence of negative behaviour. Moinard et al. (2003) identified artificial compared to natural lighting as a risk factor for tail biting in both the farrowing and nursery units [11].

The aforementioned studies show no real consensus. Differences in study design such as different photoperiods, lack of a control group, use of one gender, low numbers, or a lack of information regarding the kind or intensity of light that was used complicate their interpretation. However, optimizing lighting in the nursery unit might still be an easy and cost-effective way to improve the health and performance parameters of piglets, without the use of antibiotics. The present study aimed to fill the gaps in the current knowledge regarding the impact of different photoperiods on the performance and health, parameters, behaviour, and welfare of nursery piglets.

## 2. Materials and Methods

### 2.1. Animals and Location

The study was conducted on the pig farm of Flanders Research Institute for Agriculture, Fisheries and Food (ILVO) in Melle, Belgium. There were two batches of 144 piglets (Rattlerow-Seghers × Belgian Piétrain) that were weaned at four weeks of age, one batch in October and one in November 2020. Both SHORT and LONG piglets had a mean weaning weight of 8.1 kg.

In each treatment group, 144 piglets, housed in 24 pens distributed in 4 compartments were included. The piglets were monitored for five weeks, i.e., from weaning until the end of the nursery period. There were 8 pens in each compartment, with a surface of 1 m × 1.8 m per pen. Only 6 out of 8 pens from each compartment were included in the study. Any pigs in the 2 remaining pens were not monitored for this study. Each pen housed 6 piglets of the same gender (gilts or barrows). Piglets were allocated to a pen based on weaning weight and gender. Additionally, littermates were kept separately. Tails were docked and the boars were castrated in the farrowing house at three days of age.

### 2.2. Lighting

In the farrowing unit, the piglets were exposed 8 h of artificial light, in addition to a natural photoperiod. To avoid interference of natural light, the windows of the nursery units were covered with a thick, black, plastic sheet. The piglets were exposed to either a short photoperiod of 8 h light and 16 h dark (SHORT) or a long photoperiod of 16 h light and 8 h dark (LONG) for the entire nursery period. Treatments were assigned randomly. Four compartments were exposed to the SHORT and another 4 to the LONG photoperiod.

Lighting was provided using 4 fluorescent tubes (49 Watt, 4900 lumen) per compartment. To measure the light intensity (lux), a Testo model 545, VWR (Testo SE&co. KGaA, Titisee-Neustadt, Germany) was used at 5 points in every pen, before piglet allocation: left, middle, and right sides of the feeding through (FT); in the centre of the pen (C); and at both drinking nipples (DN). Measurements were performed at approximately pig height (30 cm from the ground). Mean values of 134 ± 29 lux in the LONG and 132 ± 29 lux in the SHORT photoperiod compartments were measured.

### 2.3. Ventilation, Nutrition, and Management

The compartments were mechanically ventilated. The temperature in the compartments was set from 27 °C at the start to 23 °C at the end of the experiment. The temperature and relative humidity were registered hourly.

Piglets were fed a starter crumble (9850 MJ/kg NE, 18.5% RE) during the first two weeks after weaning and switched to a weaning crumble (9950 MJ/kg NE, 18.0% RE) for the remainder of the nursery period. Feeders were refilled manually when the level of feed dropped below a subjective amount of feed, on average 9 times per pen. Feed and water were provided ad libitum. Enrichment was provided to promote interaction between animals from adjacent pens. This included braided cotton ropes, chains linked with plastic balls between pens, and chains connected to plastic dishes. These are ingestible, edible, movable, investigable, and manipulatable, which means they are in accordance with council directive 2008/120/EC [26] and recommendation 2016/336 of the European Commission [27].

### 2.4. Performance Parameters

The feed consumption of each pen was determined daily during the first week and weekly thereafter. The feeding throughs were weighed and compared to the tare weight of the troughs. The difference was used to calculate the average daily feed intake at pen level (ADFI). During the first week, this was calculated by dividing the daily feed disappearance by the number of animals. During the remaining weeks, the feed disappearance was divided by the number of animals in the pen times seven.

Piglets were weighed individually at the start of the experiment and weekly afterwards to determine the average daily gain (ADG) and feed conversion ratio (FCR) for the whole period.

### 2.5. Faecal Consistency

A faecal consistency score (FCS) was obtained for each pen using a tagged visual analogue scale (t-VAS scale). Each pen was given a score between 0 (normal faeces) and 100 (severe watery diarrhoea). The scores corresponded to the following observations of the faeces: 0 solid, 20 moderately solid, 40 moist, 60 mild diarrhoea, 80 yellow, watery diarrhoea, and 100 severe, watery diarrhoea. An experienced observer attributed the scores based on the general look of all the feces in the pen. Scoring was performed twice a week and was performed by the same person. Given the specific experimental design, blinding could not be guaranteed.

### 2.6. Animal Behaviour and Welfare

The main behaviours of all the piglets of each pen were recorded such as eating and drinking, dunging, resting, moving, manipulating, interacting with pen-mates, playing, mounting, ear and tail biting, and aggressive behaviour. The number of pigs exhibiting a certain behaviour were counted for each pen in a compartment (going clockwise for pen 1 to 6). This method was repeated nine more times, moving from pen to pen in the same order (scan sampling). All four compartments were scored in the same way, moving clockwise from compartment to compartment. Pens and compartments were scored in the same order each time. During the first replicate, the pigs were rated between 8 and 11 a.m. and between 12 a.m. and 15 p.m. during the second replicate. The following welfare indicators were scored:Tail biting (0: no lesions, 4: tail completely chewed off, or recovered),Ear biting (0: no lesions, 4: most of the ear has been bitten off or severe bitemarks on both ear),Skin lesions (0: no lesions, 5: multiple lesions on multiple body parts),Abdominal distension (0: empty abdomen, 1: distended abdomen).

Behaviour and welfare parameters were scored daily during the first week after weaning and weekly afterwards. For each compartment, welfare scoring was always performed first, to allow the piglets to become accustomed to the observer before scoring the behaviour. Observations and scorings were always performed by the same person. This person did not perform the FCS and was blind to the study design.

### 2.7. Statistical Analysis

Statistical analysis was performed in the software package R (version 4.1.1) using the lme4 package [28]. For the statistical analyses of pig live performance data, pen was considered as experimental unit. A linear mixed model was used with treatment as a fixed effect and weight at the start as co-variable and pen within replicate as a random effect for the entire nursery period. For the daily and weekly performance data, a linear mixed model was used with treatment, time, and their interaction as fixed effects and weight at the start as co-variable and pen within replicate as random effects for the entire nursery period.

Fecal consistency, behaviour, and welfare indicators were analyzed using a linear mixed model with treatment and time as fixed factors and replicate as a random factor. When the interaction was non-significant (*p* > 0.05), it was removed. The results were considered significant when *p*-values were lower than 0.05. Tukey’s post hoc test was used to compare treatment means.

## 3. Results

### 3.1. Performance Parameters

The ADG during the entire nursery period tended to be lower for the LONG group (385 ± 79 g) than in the SHORT group (403 ± 82 g) (*p* = 0.063) (Table 1). The same trend could be observed for the average weight of the piglets at the end of the nursery period, which was 21.6 ± 1.4 kg in the LONG group and 22.2 ± 1.6 kg in the SHORT group (*p* = 0.064). Overall, ADFI (0.496 vs. 0.510, *p* = 0.136) and FCR (1.29 vs. 1.26, *p* = 0.204) did not differ significantly between treatment groups (Table 1). The results for daily feed intake at daily basis for the first week (*p* treatment = 0.507, *p* day < 0.001) and on a weekly basis for the entire nursery period (*p* treatment = 0.136, *p* week < 0.001) did also not differ significantly (Figure 1). Weekly evaluation of daily gain showed a tendency for interaction between treatment and week (*p* treatment × *p* week = 0.096, *p* treatment = 0.602, *p* week ≤ 0.001) (Figure 2). Post hoc analyses did not, however, indicate significant differences for any of the weekly results (*p* treatment = 0.064, *p* time < 0.001).

### 3.2. Faecal Consistency

Time and treatment showed an interaction for FCS (*p* Time × treatment ≤ 0.005; *p* time < 0.001, *p* treatment < 0.001). Post hoc results indicated that the LONG piglets showed significantly higher FCS on days 7 (LONG: 21.9 ± 21.4 vs. SHORT: 10.5 ± 13.8), 11 (LONG: 40.3 ± 30.0 vs. SHORT: 14.3 ± 19.6), 21 (LONG: 20.6 ± 17.5 vs. SHORT: 7.8 ± 9.8), and 25 (LONG: 26.3 ± 22.0 vs. SHORT: 11.3 ± 13.6) (Figure 3). Diarrhoea (FCS > 60) was observed at only one timepoint (day 4 after weaning) in one out of 24 pens for the SHORT group, whereas the LONG group showed diarrhoea at five timepoints (day 4, 8, 11, 19, and 25) with a peak on day 11 (2, 2, 9, and 1 pen(s)) (Figure 4). Only one LONG piglet (5 days post-weaning, pen A15) died during the follow up.

### 3.3. Animal Behaviour and Welfare

There were no significant differences in behaviour overall (Figure 5). During the first seven days, significant differences were observed across time for resting, movement, and playing behaviour (data not shown). A tendency towards more aggressive behaviour was seen in the long photoperiod group during the first seven days of the nursery period (*p* = 0.071). There were no significant differences in the mean skin lesion score (1.6 ± 1.3 vs. 1.5 ± 1.3), prevalence of tail lesions (2.8 ± 18.4% vs. 2.3 ± 18.7%), or in abdominal filling (80.9 ± 0.4 vs. 80.4 ± 0.4) between the SHORT and LONG groups and none of the piglets showed ear biting lesions (*p* > 0.05).

## 4. Discussion

The present study could not demonstrate beneficial effects of a LONG photoperiod on the performance, health, and welfare of pigs during the nursery period.

Contrary to these results, some studies showed that prolonging the photoperiod increased ADFI, especially in the first weeks after weaning [6,7]. These studies suggest that the increased feed visibility might allow more piglets to eat. The longer photoperiod would also provide more eating opportunities for smaller and weaker piglets. Additionally, piglets are social animals, meaning that piglets use the feeder more often if other piglets are also eating. However, an increase in photoperiod may also increase activity after weaning, with piglets being more inclined to discover their environment, instead of resting or eating, thus increasing energy expenditure [13,29]. A numerical increase in playing and moving behaviour and a numerical decrease in eating and drinking behaviour was observed in the LONG compared to the SHORT piglets. Furthermore, ADFI was slightly, numerically lower in the LONG animals (*p* > 0.05). It could be hypothesized that the higher energy expenditure was not compensated by an increase in ADFI.

Nielsen et al. (1995) observed that pigs that were housed in larger groups, with a single feeding space, showed fewer and shorter feeding bouts [30]. The increased competition for the feeder could have resulted in aggression and increased stress. A prolonged photoperiod could be a similar stressor, inducing more aggressive behaviour, leaving less time to feed, and increasing stress. However, the results of the present study do not indicate relevant differences in aggressive behaviour nor in the skin lesion score. The tail lesion scores were also not significantly different between the LONG and SHORT piglets, leading to the abandonment of this hypothesis.

Next to improved performance, an improvement of feeding intake was anticipated shortly after weaning, leading to a lower prevalence of diarrhoea. However, this could not be confirmed by our results. The FCS and the number of pens with diarrhoea (FCS > 60) was numerically higher in the LONG compared to the SHORT pigs. This was particularly observed during the first weeks after weaning. During the first weeks after weaning, piglets’ digestive tracts are still adapting to a change in diet. This possibly leads to dysbacteriosis, possibly resulting in diarrhoea [31]. However, in this study the cause for the change in FCS between the LONG and SHORT piglets remains unclear. The increase in FCS could explain the lower ADG of the LONG piglets, which could lead to a lower end bodyweight. These findings indicate that increasing the photoperiod may not improve but could instead hamper the health of weaned piglets. Future research should evaluate the repeatability of these results and delve deeper into the mechanisms behind the increased FCS.

The study was performed on a single farm, with farm-specific conditions. Therefore, different conditions for lighting, housing, management, feeding, and/or drinking water may lead to different results. Furthermore, blinding could not be ensured for the fecal consistency scoring. Finally, the study was performed during autumn (October to December). Consequently, the influence of seasonality cannot be excluded. However, the study was performed under stocking densities that are similar to commercial farms, two replicates, and a reasonably large number of animals (288 piglets in total) were included. Finally, a large number of different variables related to performance, health, behaviour, and welfare were included in this study.

## 5. Conclusions

Under the present conditions, extending the photoperiod from 8 to 16 h of light per day did not significantly influence the performance, health, and welfare of nursery piglets. Furthermore, this extension negatively affected the FCS shortly after weaning. Therefore, the ‘beneficial’ effects of keeping the lights on shortly after weaning, could not be demonstrated, leaving this practice to be questioned. The potential of light as a way to increase feed intake after weaning remains to be determined. Further research should aim to optimize the lighting schedules that are used in the nursery. Standardization of the methods that are employed in this field of study might further improve the transferability of future studies.

## Figures and Tables

**Figure 1 vetsci-10-00137-f001:**
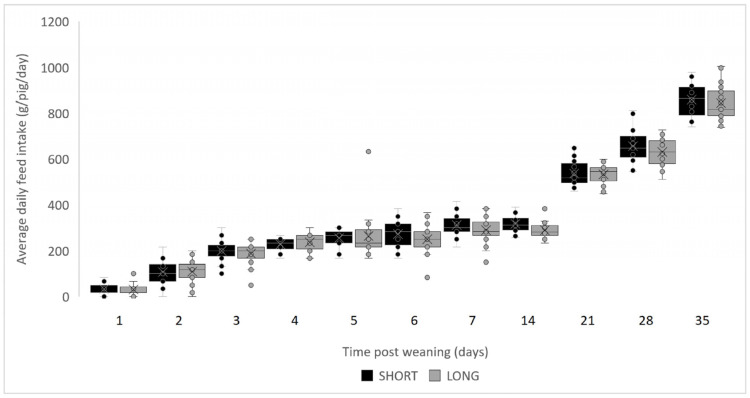
Average daily feed intake for SHORT (black, 8 h light and 16 h dark) and LONG (grey, 16 h light and 8 h dark) groups, daily for the first week (*p* treatment = 0.507, *p* day < 0.001) and weekly afterwards (*p* treatment = 0.136, *p* week < 0.001). Boxplot with jitter for each of the timepoints (1, 2, 3, 4, 5, 6, 7, 14, 21, 28, and 35 days post-weaning). The ADFI is the difference between the obtained and the tare weight of the feeding through divided by the number of pigs. For timepoints 14, 21, 28, and 35, this measurement was additionally divided by the number of days between weighings. Pen data are provided for each timepoint (circles). Median values are given by the horizontal line and the means are indicated by the “X”. The first and third interquartile ranges are given in by the lower and upper border of the boxplot. The minimum and maximum values are given by the whiskers. Outliers are shown outside of the whiskers.

**Figure 2 vetsci-10-00137-f002:**
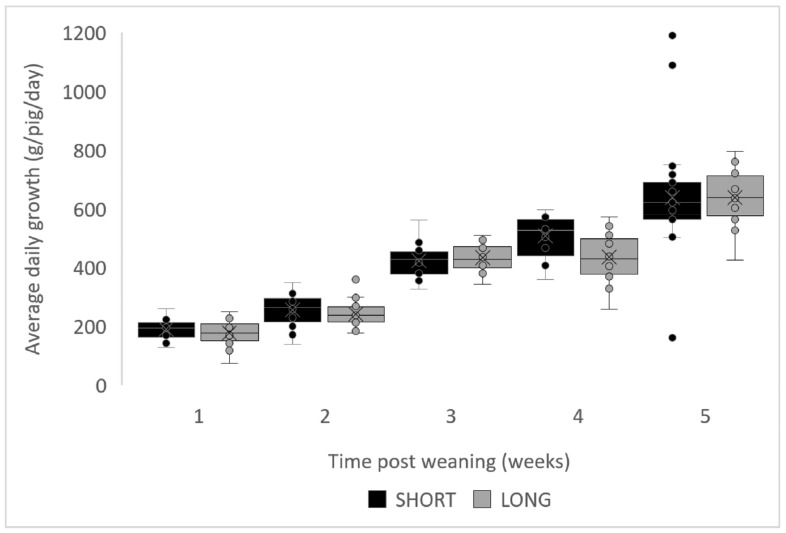
Average daily gain for SHORT (black, 8 h light and 16 h dark) and LONG (grey, 16 h light and 8 h dark) groups (*p* treatment × *p* week = 0.096, P treatment = 0.602, *p* week ≤ 0.001)**.** Boxplot with jitter for each of the weekly (1–5 weeks post-weaning) timepoints. Piglets of each pen were weighed at the end of every week. The average weight was calculated for each pen. This weight was subtracted from the previous average (or starting) weight and divided by 42 (6 pigs, 7 days) to get the ADG. Pen data are provided for each timepoint (circles). Median values are given by the horizontal line and means are indicated by the “X”. The first and third interquartile ranges are given in by the lower and upper border of the boxplot. The minimum and maximum values are given by the whiskers. Outliers are shown outside of the whiskers.

**Figure 3 vetsci-10-00137-f003:**
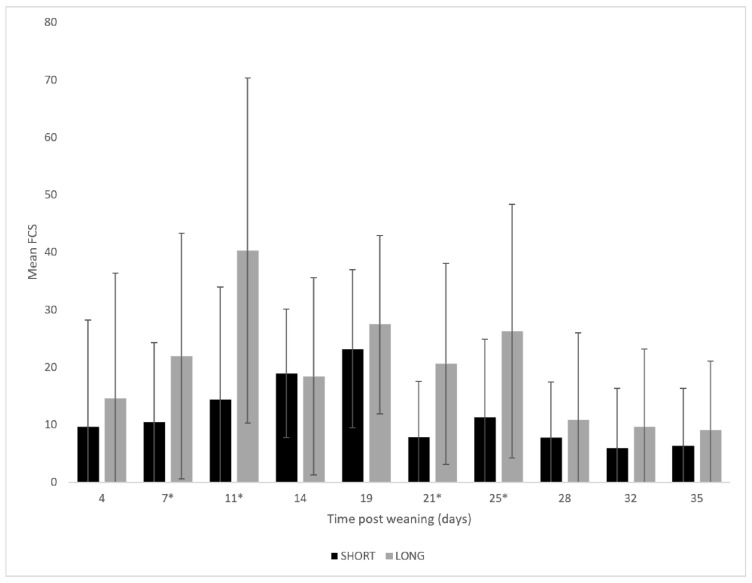
Faecal consistency scores (FCS) of piglets in the SHORT (black, 8 h light 16 h dark) and LONG groups (grey, 16 h light and 8 h dark). Scores (range: 0 = normal faeces) to 100 = severe watery diarrhoea) were measured twice weekly for each pen during the whole nursery period. Mean values are given for all the study pens (24 pens of 6 piglets for short and long groups). The standard deviation is given by the whiskers. (*p* treatment × time < 0.001, *p* time < 0.001, *p* treatment < 0.001; * indicates significant differences per time point).

**Figure 4 vetsci-10-00137-f004:**
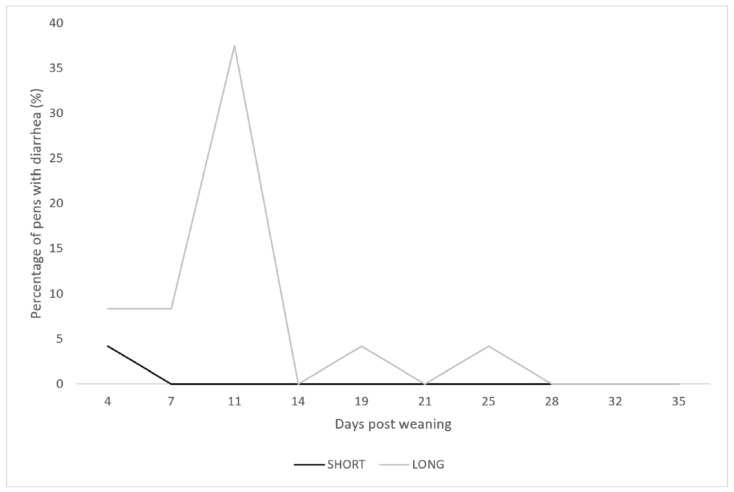
Percentage of pens with diarrhoea for SHORT (black, 8 h light and 16 h dark) compared to LONG photoperiod piglets (grey, 16 h light and 8 h dark).

**Figure 5 vetsci-10-00137-f005:**
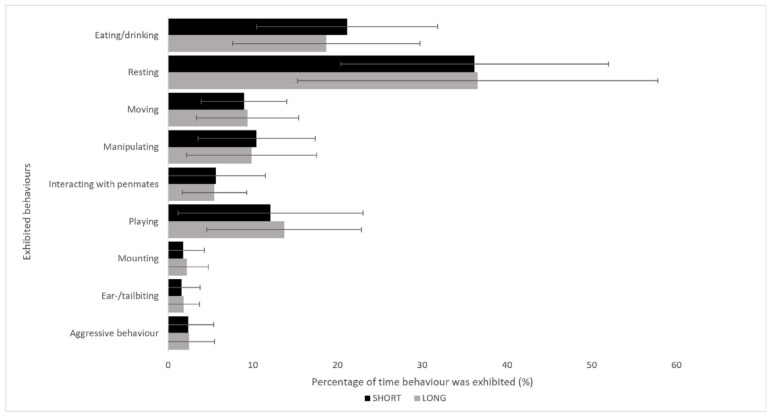
Behavioural parameters of SHORT photoperiod piglets (black, 8 h light and 16 h dark) compared to LONG photoperiod piglets (grey, 16 h light and 8 h dark). This graph represents the mean values for the times (percentage) that the behaviour was observed over the different scoring timeframes. The findings were not significant between treatments (*p* > 0.05). Whiskers represent the standard error of each mean value.

**Table 1 vetsci-10-00137-t001:** Performance parameters of short photoperiod piglets (SHORT, 8 h light and 16 h dark) compared to long photoperiod piglets (LONG, 16 h light and 8 h dark). Mean values and their standard deviations are given for each parameter. The F-value of the starting weight was not determined. Therefore, no statistical analysis was made for this parameter (not available or NA).

Parameter	SHORT Mean ± SD	LONG Mean ± SD	Df	F-Value	*p*-Value
BW_start_ (kg)	8.1 ± 1.10	8.1 ± 1.15	1	NA	0.743
BW_end_ (kg)	22.19 ± 1.63	21.59 ± 1.37	1	3.63	0.064
ADFI (kg)	0.510 ± 0.041	0.496 ± 0.036	1	2.31	0.136
FCR (kg/kg)	1.26 ± 0.090	1.29 ± 0.057	1	1.66	0.204
ADG (kg)	0.403 ± 0.082	0.385 ± 0.079	1	3.63	0.063

SHORT: eight hours of light and 16 h of darkness (SHORT), LONG: 16 h of light and 8 h of darkness (LONG), SD: standard deviation, NA: not available, ADFI: mean average daily feed intake, BW_start_: mean bodyweight of piglets at the start of the trial, BW_end_: mean bodyweight of piglets at the end of the trial, FCR: feed conversion ratio, ADG: average daily gain.

## Data Availability

The raw data of this publication can be made available by the authors upon request.

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
