# Peer review of "Effect of Extended Photoperiod on Performance, Health, and Behavioural Parameters in Nursery Pigs"

_vetsci, 2023, doi:10.3390/vetsci10020137_

Round 1
Reviewer 1 Report
General comment
Thank you for this nice study and manuscript, I believe this is a topic of interest for the pig production science. The quality of the manuscript must be improved to meet the requirements for publications. I recommend that the authors add information about the studies they refer to in the introduction, give more details about their methods, and provide their full results (some statistical information is missing). See below my specific comments.
Introduction
L51: Facilitates
L54: Add “i.e. environmental factors” after “Zeitgeber”
L75: Add reference number, otherwise it is difficult to find it in the references section
L77: Add reference number, and give details of the length of photoperiod and weaning age in the cited study
L79: Add reference number, and give details of the length of photoperiod, or detail why the LED-light was important (refer to the hypothesis of the cited study)
L83-84: You wrote that “multiple studies found no, or a negative effect of prolonging photoperiod” but the first example cited after that is actually about positive results… You need to provide more examples of negative results and detail a bit more the conditions of the cited studies, so the reader can make a comparison with the studies cited above (which also need to be more detailed).
L85: Add details about the lighting conditions of the farrowing house.
L86-88: As mentioned above, this is a positive result of prolonging photoperiod – you also need to link the two sentences together for better reading, e.g. “…in the farrowing unit, those piglets…”
L88: “Photoperiod may also…” this statement is elusive, I guess you are referring to “longer/extended photoperiod”?
L89: Add reference number. In this example, you compare “natural” vs “artificial” lightning, so the photoperiods were the same but the quality of the light differed ? You need to explicitly say that, since in the previous examples you focused on the length of the photoperiod (which is also the scope of your paper).
L90: Add reference number
Material and Methods
L122-126: Please give average values of light intensity here, I saw them mentioned in the results but I believe they are not a result of your treatment (this is not a difference you hoped to create) but rather a condition (you made sure that the intensity was the same in both treatments so only the length of the photoperiod would be a factor of variation).
L138-141: How often was the feeder refilled? Was it done manually or automatically? How much feed did you refilled each time (in average)? Reading this paragraph, I would understand that you filled them up once a week, when weighing them.
L148-154: I need more details on how this scoring was performed: did you score each faeces present in the pen and then averaged it, or did you give a general score for the pen?
Did you perform a validation of the scoring method before using it (e.g. inter-rater agreement)? If yes, please provide the way this was done and the agreement score obtained.
I am (in general) cautious with visual analogous scale scoring, especially if the observer is not blinded to the treatments.
L153: What exactly prevented the rater to be blinded to the treatments? I do not understand why it was possible to blind the behaviour observer but not the FC rater?
L160: Repeated 10 times over which period? When (times of the day) and how often were the pigs observed?
L161: So you had a time gap between compartments. Did you keep the same order throughout the experimental period or did you (randomly?) rotate the compartment order? This has implications for your research and the repeatability of your results (which you mention as an advice to future researcher at the end of the discussion… so you need to provide ALL information that would help reproduce your research!)
L168: What are your welfare measures? In this section only behaviours are listed and they are not purely measures of welfare (their outcome could inform on the welfare status of the pigs, but you do not explain which behaviours are indicators of welfare for your study). In general, welfare is considered to be the association of behaviour, health and physiological factors – it is not a single measure but a concept that arise from multiple indicators.
L173: I find the statistical section quite poorly detailed… which packages in R did you use? Did you check the normality of the data, and/or did you have to transform it? In linear mixed models (in R at least), random slopes and random intercepts can be included, but here you only mention the random intercepts. Why was faecal consistency analysed differently?
Results
L182-183: As mentioned above, this should be in the Material and Methods section
L190: space missing after “group(P=0.064)”.
L185-192: Please add the statistical outcomes (t/F-value, P-value) for those results (it is not possible to guess them from the figures)
L241-248: Please add the statistical outcomes (t/F-value, P-value) for those results (it is not possible to guess them from the figures)
L249: As you wrote yourself, please mention the timepoint and group for the dead piglet.
L263: “based on the observation timepoints” - Do you mean that there were some differences across time?
L266: please provide the estimated means and SE for each treatment for your result on aggressive behaviour, or the difference between the treatments, so that the reader can appreciate the biological relevance of this difference
Tables and Figures
Table 1: Please also add the t/F-value and degrees of freedom
Figure 1 (A&B): I guess that you divided the through weight difference by the number of days between two weighings (otherwise it is not a "daily" feed intake, but a weekly feed intake) ?
Figure 4: I am not sure that this choice of representation is wise… usually the timeline is represented on the x-axis because it is the independent variable along which the response variable (FCS) changes.
Figures 5 and 6: I am surprised not to see SE on those graphs, are those the outcomes of your statistical model or are they the raw data?
Table 2: The header says “mean±SD” but no SD is mentioned in the table
Discussion
L281: I am not sure that “respond to peers” is the best way to describe the propensity of pigs to synchronise their activities. Because it is not a response but a choice, mostly due to their behavioural ecology as a gregarious/social species.
L284: but one could easily argue that there could be more eating as well since pigs naturally do not feed during the nocturnal period.
L286: Add “compared to SHORT piglets” after “observed in LONG piglets”
L289: You need to discuss the hypotheses that you developed in the introduction, not make them in the discussion – this one is CENTRAL to your research and should have been more developed in the introduction (with supporting citations, etc.)
L294-295: High feed intake can also be responsible for diarrhoea, as the pigs are still adapting to the solid diet.
L296: this was numerically higher (I did not see statistics done on this parameter)
L299: It is grammatically incorrect to start a sentence with "which", replace “.” with “,”.
L301: Add “instead” between “could” and “hamper”
L301-303: That statement sounds strange to me: you criticised the lack of agreement between studies in the introduction (the lack of consistency in the experimental set-ups etc.) and in the next paragraph you mention that the results could be different if the experiment was repeated on a different farm (i.e. with different conditions). What did you do to encourage/improve the repeatability of your study? Also, you advise to research the mechanisms behind the increase FCS, but this is a method that (from my understanding) you made up (I do not see reference to previous publications) and did not give much details about how to perform the scoring (how many samples per pen? How can one differentiate “moderately solid” and “moist”? …). Additionally, one major flaw of your assessment is that the (unique) rater was not blinded to the treatments.
L306: “may lead to different results” instead of “alter results” – your results are not going to change, but someone can find different results.
L310: How can you affirm that the study had “a high power”? Did you perform power analysis on your models (or before the experiment, to determine sample size)? What was the effect size of your treatment (this is the main method to assess the power of your study and analyses)?
L314: You did not measure nor discussed welfare outcomes for the pigs.
L319: What do you mean with “Standardisation of the methods”? If your evaluation method were not standardised or validated, you should not have used them (or used them conjointly with a validated standard measure to be able to confirm the reliability of your methods).
Other sections
L335-339: This is the general statement of MDPI...
Author Response
Dear reviewer,
Thank you very much for your extensive review of our publication. Please find a response for each of your comments below.
L51: Facilitates
This has been revised as suggested.
L54: Add “i.e. environmental factors” after “Zeitgeber”
This has been revised as suggested.
L75: Add reference number, otherwise it is difficult to find it in the references section
This has been done as suggested.
L77: Add reference number, and give details of the length of photoperiod and weaning age in the cited study
This has been revised as follows: "Niekamp et al (2007) showed that an increased photoperiod (16L8D) and increasing the weaning age (28 vs. 14 days of age) improved ADG, end weight and several immunity parameters [7].".
L79: Add reference number, and give details of the length of photoperiod, or detail why the LED-light was important (refer to the hypothesis of the cited study)
To clear up this section it was rewritten as follows: "Kluivers-Poodt et al (2018) exposed piglets, kept under either TL- and LED-lighting, to an extended photoperiod (16L8D) [8]. Results showed that the feed conversion ratio (FCR) and ADFI of piglets kept under LED-light were higher during the first 14 days, compared to piglets kept under TL-light. However, alternating the lighting schedule, by switching between high (150 lux) and low (40 lux) intensities throughout the day, improved FCR and ADG compared to the TL-group [8]"
L83-84: You wrote that “multiple studies found no, or a negative effect of prolonging photoperiod” but the first example cited after that is actually about positive results… You need to provide more examples of negative results and detail a bit more the conditions of the cited studies, so the reader can make a comparison with the studies cited above (which also need to be more detailed).
This is not the case, please see the answer to your comment of line 86-88.
L85: Add details about the lighting conditions of the farrowing house.
No details were given about the lighting conditions of the farrowing house in this publication.
L86-88: As mentioned above, this is a positive result of prolonging photoperiod – you also need to link the two sentences together for better reading, e.g. “…in the farrowing unit, those piglets…”
To my opinion this is not a positive result. Piglet weight and feed conversion rate remained the same, while feed intake increased. There are multiple possible explanations for this. 1) More feed was wasted due to increased restlessness and playing with feed 2) More feed was consumed, but did not result in weight gain possibly due to increased aggressive behavior resulting in more stress, and decreasing metabolism efficiency. In the end the result is unfavorable due to decreased welfare of the pigs and higher feeding cost. Further inspection of the results also showed a numerical increase in the feed intake and feed conversion ratio during the first week after weaning. This hints at increased hierarchy fighting during the first week. This likely also depends on the manner of mixing at weaning and the differences (or absence thereof) in weaning weights of pen mates.
L88: “Photoperiod may also…” this statement is elusive, I guess you are referring to “longer/extended photoperiod”?
This was revised to clear up confusion: "Differences in photoperiod may also affect behaviour and welfare."
L89: Add reference number. In this example, you compare “natural” vs “artificial” lightning, so the photoperiods were the same but the quality of the light differed ? You need to explicitly say that, since in the previous examples you focused on the length of the photoperiod (which is also the scope of your paper).
This difference has been stressed as follows: "Lastly, the quality of light might influence the occurrence of negative behaviour. Moinard et al (2003) identified artificial, compared to natural, lighting as a risk factor for tail biting in both the farrowing and nursery units [11]."
L90: Add reference number
This has been changed as follows: "Photoperiod may also affect behaviour and welfare [10, 11]."
Material and Methods
L122-126: Please give average values of light intensity here, I saw them mentioned in the results but I believe they are not a result of your treatment (this is not a difference you hoped to create) but rather a condition (you made sure that the intensity was the same in both treatments so only the length of the photoperiod would be a factor of variation).
The following sentence has been added: "Mean values of 134±29 lux in the LONG and 132±29 lux in the SHORT photoperiod compartments were measured."
L138-141: How often was the feeder refilled? Was it done manually or automatically? How much feed did you refilled each time (in average)? Reading this paragraph, I would understand that you filled them up once a week, when weighing them.
Feeders were refilled manually. This was done based on the remaining volume of feed inside the feeder. Also, feeders were not necessarily filled for each pen on the same day. The following sentence was added to clarify. "Feeders were filled manually. This happened when the amount of feed left in the feeder was visually too low, on average 9 times per pen."
L148-154: I need more details on how this scoring was performed: did you score each faeces present in the pen and then averaged it, or did you give a general score for the pen?
Did you perform a validation of the scoring method before using it (e.g. inter-rater agreement)? If yes, please provide the way this was done and the agreement score obtained.
I am (in general) cautious with visual analogous scale scoring, especially if the observer is not blinded to the treatments.
A general feces score was given for each pen. This method of faecal consistency scoring is routinely performed by this rater in this case. Raters are trained in this scoring system before applying it in an experimental setting. Additionally, the scoring form shows photographs depicting the specific fecal consistencies. The following sentence has been added to clear this up. "An experienced observer attributed the scores were attributed based on the general look of all feces in the pen.".
L153: What exactly prevented the rater to be blinded to the treatments? I do not understand why it was possible to blind the behaviour observer but not the FC rater?
The person assessing the behaviour (rater r1) did not know the lighting schedules used in the different compartments. The lighting was turned on and off on a timer. Behavioural scoring was performed after the lights turned on and before they were turned back off. The rater scoring the fecal consistency (rater r2) programmed the timers. So while r1 was blind to the treatment, r2 was not.
L160: Repeated 10 times over which period? When (times of the day) and how often were the pigs observed?
During the first replicate, the pigs were rated between 8 and 11 a.m. and between 12 a.m. and 15 p.m. during the second replicate. The behavioural rater checked the pigs once every day during the first week, and once a week for the subsequent weeks. The pigs were checked daily by farm personnel. The number of animals exhibiting a certain behaviour at that time were written down. Next, I moved on to the next pen to note down the number of animals exhibiting a certain behaviour. After all six pens had been scored once, the scoring was repeated 9 more times in the same way, moving from pen to pen. This resulted in the pigs being observed 10 times per compartment. This took around 30 minutes per pen.
L161: So you had a time gap between compartments. Did you keep the same order throughout the experimental period or did you (randomly?) rotate the compartment order? This has implications for your research and the repeatability of your results (which you mention as an advice to future researcher at the end of the discussion… so you need to provide ALL information that would help reproduce your research!)
The pigs and the compartments were scored in the same order throughout the study. As mentioned before the time of day differed between replicates. The section was rewritten, hopefully clearing up the confusion. "The number of pigs exhibiting a certain behaviour were counted for each pen in a compartment (going clockwise from pen 1 to 6). This method was repeated nine more times, moving from pen to pen in the same order (scan sampling). All four compartments were scored in the same way, moving from compartment to compartment in a clockwise fashion. Pens and compartments were scored in the same order each time. During the first replicate the pigs were rated between 8 and 11 a.m. and between 12 a.m. and 15 p.m. during the second replicate."
L168: What are your welfare measures? In this section only behaviours are listed and they are not purely measures of welfare (their outcome could inform on the welfare status of the pigs, but you do not explain which behaviours are indicators of welfare for your study). In general, welfare is considered to be the association of behaviour, health and physiological factors – it is not a single measure but a concept that arise from multiple indicators.
We agree with this remark. For this study, we decided to focus on a selection of indicators for unwanted behaviour: ear-, tail- and skin lesions; as well as abdominal filling to assess the impact of weaning stress (uptake of feed after weaning). Though they do not constitute welfare in general, significant differences between treatment groups (SHORT vs. LONG) for these indicators could hint at a welfare problem. Excessive lesions or reduced abdominal filling are also the first indications for farmers that the welfare of their pigs is compromised.
L173: I find the statistical section quite poorly detailed… which packages in R did you use? Did you check the normality of the data, and/or did you have to transform it? In linear mixed models (in R at least), random slopes and random intercepts can be included, but here you only mention the random intercepts. Why was faecal consistency analysed differently?
Statistical analysis was performed in the software package R (version 4.1.1) using the lme4 package (Bates et al., 2015). Weight, ADG, ADFI and FCR were analysed using a linear mixed model. Phase, feed, gender, treatment and their interaction and weight at the start were used as fixed factors whereas pen and replicate were used as random factors. Faecal consistency was analysed using a linear mixed model with treatment and time as fixed factors and pen as random factor. When the interaction was non-significant (P>0.05) it was removed. Results were considered significant when P-values were lower than 0.05.
- Ref: R Core Team. R: A Language and Environment for Statistical Computing; R Foundation for Statistical Computing: Vienna, Austria, 2020.
- Bates, D., Maechler, M., Bolker, B., Walker, S., 2015. {lme4}: Linear mixed-effects models using {Eigen} and {S4}.
Results
L182-183: As mentioned above, this should be in the Material and Methods section
This section has been moved to the M&M as suggested.
L190: space missing after “group(P=0.064)”.
This was adapted as suggested.
L185-192: Please add the statistical outcomes (t/F-value, P-value) for those results (it is not possible to guess them from the figures)
Results for the overall nursery period:
## Response: daily gain
## F Df Df.res Pr(>F)
## (Intercept) 31.6883 1 8.328 0.0004268 ***
## behandeling1 3.6352 1 41.015 0.0635905 .
## sex 1.1075 1 41.007 0.2987908
## gstart 0.1155 1 8.719 0.7420042
## Response: daily feed intake
## F Df Df.res Pr(>F)
## (Intercept) 24.3493 1 8.328 0.001016 **
## behandeling1 2.3079 1 41.015 0.136389
## sex 0.0004 1 41.007 0.984886
## gstart 2.7761 1 8.719 0.131120
## Response: vctot
## F Df Df.res Pr(>F)
## (Intercept) 44.4731 1 8.328 0.0001313 ***
## behandeling1 1.6651 1 41.015 0.2041479
## sex 4.5320 1 41.007 0.0393212 *
## gstart 9.8712 1 8.719 0.0123653 *
## ---
## Response: end weight
## F Df Df.res Pr(>F)
## (Intercept) 31.6883 1 8.328 0.0004268 ***
## behandeling1 3.6352 1 41.015 0.0635905 .
## sex 1.1075 1 41.007 0.2987908
## gstart 7.0164 1 8.719 0.0272365 *
L241-248: Please add the statistical outcomes (t/F-value, P-value) for those results (it is not possible to guess them from the figures) A table was added with the different statistical values for figure 4.
We have included p-values for all relevant values. P-values were not given for diarrhoea scores. These scores are the numerical values of the number of pens exhibiting diarrhoea based on the cut off value of an FCS of 60. These are not multiple different values that can be compared. Therefore, no statistical parameters (t/F or p-values) are given for these scores. Therefore, we did not alter the text. Additionally, because of the low number of pens with diarrhoea, our model did not run.
L249: As you wrote yourself, please mention the timepoint and group for the dead piglet.
This has been added: "Only one LONG piglet (5 days post weaning, pen A15) died during the follow up.".
L263: “based on the observation timepoints” - Do you mean that there were some differences across time?
This has been adapted as follows: "During the first seven days, significant differences were observed across time for resting, movement and playing behaviours (data not shown).".
L266: please provide the estimated means and SE for each treatment for your result on aggressive behaviour, or the difference between the treatments, so that the reader can appreciate the biological relevance of this difference
This has been adapted in the text. Consequently the table was removed in order to avoid excessive repetition in the manuscript.
Tables and Figures
Table 1: Please also add the t/F-value and degrees of freedom
We have added the requested data to the table.
Figure 1 (A&B): I guess that you divided the through weight difference by the number of days between two weighings (otherwise it is not a "daily" feed intake, but a weekly feed intake)?
This is correct. The following section was added in the materials and methods to clear this up. "During the first week this was calculated by dividing the daily feed disappearance by the number of animals. During the remaining weeks the feed disappearance was divided by the number of animals in the pen times seven.". This was additionally indicated underneath the figures by the following sentence: " For timepoints 14, 21, 28 and 35 this measurement was additionally divided by the number of days between weighings ".
Figure 4: I am not sure that this choice of representation is wise… usually the timeline is represented on the x-axis because it is the independent variable along which the response variable (FCS) changes.
The figure has been changed as suggested.
Figures 5 and 6: I am surprised not to see SE on those graphs, are those the outcomes of your statistical model or are they the raw data?
Figure 5 is raw pen data. Pens were classified showing signs of diarrhoea (FCS>60) or not (FCS<60). Therefore, these are absolute values, and no standard errors could be added. Standard errors and a y-axis were added for Figure 6. This way the results can still be interpreted, but the standard errors are given.
Table 2: The header says “mean±SD” but no SD is mentioned in the table
This table was removed in order to avoid repetition.
Discussion
L281: I am not sure that “respond to peers” is the best way to describe the propensity of pigs to synchronise their activities. Because it is not a response but a choice, mostly due to their behavioural ecology as a gregarious/social species.
This sentence has been revised into: "Additionally, piglets are social animals, meaning that piglets use the feeder more often if other piglets are also eating.".
L284: but one could easily argue that there could be more eating as well since pigs naturally do not feed during the nocturnal period.
This could indeed be the case, but this was not the hypothesis of the cited papers. They indicate or suggest that piglets become more restless. Furthermore, no increase in feeding behaviour was observed behaviorally (quite the opposite, our data show a numerical decrease in feeding behaviour), nor was this hypothesis reflected in the ADFI data of our study.
L286: Add “compared to SHORT piglets” after “observed in LONG piglets”
This was revised as suggested.
L289: You need to discuss the hypotheses that you developed in the introduction, not make them in the discussion – this one is CENTRAL to your research and should have been more developed in the introduction (with supporting citations, etc.)
The scientific method dictates that hypotheses need to be developed before the start of an experiment. Which we did. However, this hypothesis was formed after the experiment was concluded. It serves as a proposed hypothesis for some of the findings of our experiment. This is why it is mentioned in the discussion, rather than in the introduction. We do agree that we did not provide supporting literature. Therefore, we adapted the section as follows: “Nielsen et al (1995) observed that pigs housed in larger groups, with a single feeding space, showed less and shorter feeding bouts [27]. The increased competition for the feeder could have resulted in aggression and increased stress. A prolonged photoperiod could be a similar stressor. Inducing more aggressive behaviour, leaving less time to feed and increasing stress. However, the results of the present study do not indicate relevant differences in aggressive behaviour, nor in the skin lesion score. Also, the tail lesion scores were not significantly different between LONG and SHORT piglets. Leading to the abandonment of this hypothesis.”.
L294-295: High feed intake can also be responsible for diarrhoea, as the pigs are still adapting to the solid diet.
High feed intake could lead to an increase in FCS, but we would expect this to happen during the first week of the nursery period. The statistical differences in FCR were observed far exceeded a week. No statistical differences in feed intake were observed between the SHORT and LONG groups during this first week, nor in general.
L296: this was numerically higher (I did not see statistics done on this parameter)
"numerically" was added to indicate the absence of statistical analysis.
L299: It is grammatically incorrect to start a sentence with "which", replace “.” with “,”.
This has been revised as suggested.
L301: Add “instead” between “could” and “hamper”
This has been revised as suggested.
L301-303: That statement sounds strange to me: you criticised the lack of agreement between studies in the introduction (the lack of consistency in the experimental set-ups etc.) and in the next paragraph you mention that the results could be different if the experiment was repeated on a different farm (i.e. with different conditions). What did you do to encourage/improve the repeatability of your study? Also, you advise to research the mechanisms behind the increase FCS, but this is a method that (from my understanding) you made up (I do not see reference to previous publications) and did not give much details about how to perform the scoring (how many samples per pen? How can one differentiate “moderately solid” and “moist”? …). Additionally, one major flaw of your assessment is that the (unique) rater was not blinded to the treatments.
We agree with this comment. There is always some subjectivity involved. The scoring was performed by an experienced observer, The scoring form also depicted the different faecal consistency scores. To highlight the lack of blinding in the study we added the following sentence in this section: “Furthermore, blinding could not be ensured for the faecal consistency scoring.”.
L306: “may lead to different results” instead of “alter results” – your results are not going to change, but someone can find different results.
This was revised as suggested.
L310: How can you affirm that the study had “a high power”? Did you perform power analysis on your models (or before the experiment, to determine sample size)? What was the effect size of your treatment (this is the main method to assess the power of your study and analyses)?
In retrospect it is not correct to say that this study has a high power. We would like to refer you to the post-power calculations performed below:
Average daily growth
Power for predictor 'behandeling1', (95% confidence interval):
47.30% (44.17, 50.45)
Test: t-test
Based on 1000 simulations, (0 warnings, 0 errors)
alpha = 0.05, nrow = 46
Time elapsed: 0 h 0 m 3 s
Daily feed intake
Power for predictor 'behandeling1', (95% confidence interval):
34.80% (31.85, 37.84)
Test: t-test
Based on 1000 simulations, (0 warnings, 0 errors)
alpha = 0.05, nrow = 46
For this reason we removed this part from the discussion.
L314: You did not measure nor discussed welfare outcomes for the pigs.
As you correctly remark, we did not look at the overall welfare of the animal. However, we did score welfare indicators (tail lesions, ear lesions, skin lesions and abdominal filling). No statistically relevant differences were observed between treatments for these indicators.
L319: What do you mean with “Standardisation of the methods”? If your evaluation method were not standardised or validated, you should not have used them (or used them conjointly with a validated standard measure to be able to confirm the reliability of your methods).
This sentence was intended as a critique of the general body of literature on this subject. We have revised the sentence to clear up this misunderstanding: "Further research should aim to optimize the lighting schedules used in the nursery. Standardization of the methods employed in this field of study, might further improve the transferability of future studies."
Other sections
L335-339: This is the general statement of MDPI...
The following statement was made regarding the data availability: “The raw data of this publication can be made available by the authors upon request.”.

Reviewer 2 Report
Dear Authors,
the subject of the paper is very interesting, especially from the practical point of view. There is a lot of evidence that access to the light performed too long or when the intensity of the lighting is too high may lead to behavioural abnormalities such as high activity which have no explaation as well as the tail or ear biting. It is very valuable that Authors tested the effect of short and long photoperiod on the productive results and not only for the behaviour. On the other hand the methods of behavioural research in this project could be more effective, however there is still space in this case for the another researches and papers concerning the influence of the duration of lighting on weaners' behaviour.
I find this paper worth publishing in Veterinary Sciences but some improvements should be done as follows:
1. Please provide the weaning body weight in methods, not only in the table 1.
2. Please write if the environmental enrichment used in the farm was in agreement with Directive 2008/120/EC and Commision Recommendation (EU) 2016/336. In my opinion it is not. Please provide more details about it and explain whether you have any influence on the choice of the enrichment,
3. Line 122 "...Lighting was provided using 4 fluorescent tubes per compartment...." Could you please add more technical details?
4. Please take into consideration that longer duration of lighting may affect the eating and sleeping behaviour, especially in case of weaners which take lower places in the social hierarchy, because the normal situation is that such marginal individuals sometimes may eat only during night; when dominants and subdominants sleep. Therefore I think that your methods of behavioural observations are not perfect. Maybe you have some data (recordings) and you could add such distribution of eating, sleeping behaviour according to the position is social hierarchy.
5. It would be great if you present behavioural data concerning the duration of formation of social hierarchy in SHORT and LONG groups, because you mentioned that weaned piglets placed in pens were not littermates. If two mentioned above points are unable to correct in this paper, please think about it during planning the further research/project
6. Could you please add technical information about the device you measured the light intensity?
In my opinion your paper will be very interesting for readers, especially for pig producers and farmers and it is the main reason why it should be published.
Author Response
Dear reviewer,
Thank you for your comments. Please find our answers to your comments below.
- Please provide the weaning body weight in methods, not only in the table.
This was added in the section "animals and location".
- Please write if the environmental enrichment used in the farm was in agreement with Directive 2008/120/EC and Commision Recommendation (EU) 2016/336. In my opinion it is not. Please provide more details about it and explain whether you have any influence on the choice of the enrichment,
Enrichment promotes interaction between pens, is chewable, investigable, manipulatable, and in the case of the braided cotton ropes edible. So in that sense the enrichment is in agreement with the recommendation and the directive.
- Line 122 "...Lighting was provided using 4 fluorescent tubes per compartment...." Could you please add more technical details?
The specifications have been outlined in the M&M with the following sentence: “Lighting was provided using 4 fluorescent tubes (49 watt, 4900 lumen) per compartment.“
- Please take into consideration that longer duration of lighting may affect the eating and sleeping behaviour, especially in case of weaners which take lower places in the social hierarchy, because the normal situation is that such marginal individuals sometimes may eat only during night; when dominants and subdominants sleep. Therefore I think that your methods of behavioural observations are not perfect. Maybe you have some data (recordings) and you could add such distribution of eating, sleeping behaviour according to the position is social hierarchy.
Unfortunately, we did not study the role of social hierarchy, nor do we have video-recordings of the pens. This would indeed have interesting implications for future research.
- It would be great if you present behavioural data concerning the duration of formation of social hierarchy in SHORT and LONG groups, because you mentioned that weaned piglets placed in pens were not littermates. If two mentioned above points are unable to correct in this paper, please think about it during planning the further research/project
As mentioned before, we did not include (formation of) social hierarchy in our study. However, it is interesting to consider for the future.
- Could you please add technical information about the device you measured the light intensity?
We have specified the luxmeter in the lighting section of the M&M by the following sentence: “To measure the light intensity (lux) a Testo model 545, VWR (Testo SE&co. KGaA, Germany) was used at 5 points in every pen, before piglet allocation: left, middle and right sides of the feeding through (FT), in the centre of the pen (C) and at both drinking nipples (DN). “